# MITS: ENHANCED TREE SEARCH REASONING FOR LLMS VIA POINTWISE MUTUAL INFORMATION

## ABSTRACT

Tree search has become as a representative framework for test-time reasoning with large language models (LLMs), exemplified by methods such as Tree-of-Thought and Monte Carlo Tree Search that explore multiple reasoning paths. However, it remains difficult to provide instant and reliable quantitative assessments of intermediate reasoning step quality, and extensive path exploration is computationally costly. To address this, we propose Mutual Information Tree Search (MITS), a novel framework that guides reasoning with information-theoretic principles. MITS introduces an effective scoring function based on pointwise mutual information (PMI), which enables step-wise evaluation of reasoning paths and search tree expansion via beam search without expensive look-ahead simulations, achieving superior reasoning performances while maintaining computational efficiency. The framework is complemented by an entropy-based dynamic sampling strategy that adaptively allocates computational resources to uncertain reasoning steps where exploration is most beneficial. For final prediction, MITS employs a weighted voting scheme that combines PMI scores with prediction consensus. Through comprehensive experiments on diverse reasoning benchmarks, MITS consistently surpasses baseline methods, establishing a principled and efficient framework for LLM reasoning.

## 1 INTRODUCTION

Complex multi-step reasoning remains a fundamental challenge for Large Language Models (LLMs), particularly in tasks that require logical deduction, mathematical computation, or systematic problem-solving (Yang et al., 2025; Zhu et al., 2024; Yi et al., 2024). While Chain-of-Thought (CoT) prompting (Wei et al., 2022; Kojima et al., 2022) has emerged as a powerful technique to enhance reasoning by decomposing problems into intermediate steps, it typically generates a single reasoning path, which may lead to incorrect solutions due to error accumulation or the selection of suboptimal reasoning strategies. This limitation becomes particularly pronounced in complex reasoning tasks where multiple valid approaches exist, but only specific paths lead to correct answers.

Recent work on *inference-time scaling laws* (Snell et al., 2024; Parashar et al., 2025) reveals a promising direction: generating multiple reasoning paths can substantially improve accuracy. The key insight is that solution coverage, *i.e.*, the probability of finding at least one correct answer, scales predictably with the number of attempts. This observation has proven especially valuable in domains with verifiable answers, such as code generation and theorem proving. However, exhaustively enumerating reasoning paths is infeasible, as the search space grows exponentially with problem complexity. This leads to the central research problem: *How can we efficiently search through the vast space of possible reasoning paths to identify those most likely to yield correct solutions?*

To tackle this problem, existing approaches face several limitations. Methods based on exhaustive rollouts (Wang et al., 2023) or Monte Carlo Tree Search (MCTS; Hao et al., 2023; Qi et al., 2025; Ding et al., 2024) require extensive forward simulation, which is computationally prohibitive at scale. Self-evaluation methods struggle to provide quantitative assessments of reasoning quality, often relying on pairwise comparisons or binary judgments that fail to capture nuanced differences between paths (Xie et al., 2023; Gu et al., 2024). Most critically, existing scoring mechanisms usually favor **plausible but generic reasoning paths that could apply to many problems, rather than those specifically tailored to the given question**. These limitations highlight the need for a

principled framework that can (i) evaluate reasoning quality efficiently without costly simulations, and (ii) distinguish between generic reasoning and question-specific reasoning.

In this work, we propose Mutual Information Tree Search (MITS), a principled framework that addresses these challenges through information-theoretic guidance. Our key insight is that effective reasoning paths should exhibit high mutual information with the question, which means they should contain information that is both relevant and uniquely tied to solving the given problem. We leverage Pointwise Mutual Information (PMI) as a scoring mechanism that quantifies how much a reasoning path's plausibility increases *because of the specific question*, effectively filtering out generic or spurious reasoning patterns. Moreover, by computing scores dynamically as the reasoning process unfolds, MITS evaluates reasoning quality efficiently without expensive look-ahead simulations.

Overall, our framework introduces three technical innovations. First, we propose a **principled scoring approach** for intermediate reasoning steps using Pointwise Mutual Information (PMI), which reduces computational burden and establishes reliable criteria for trajectory selection. Second, we propose an **entropy-based dynamic sampling strategy** that adaptively allocates computational resources according to the uncertainty at each reasoning step, concentrating exploration on uncertain decision points where diverse reasoning approaches yield the greatest potential benefit. Third, we introduce a **weighted voting scheme** that combines PMI scores with prediction consensus, reducing the risk of selecting high-scoring but spurious reasoning paths by utilizing agreement across multiple reasoning trajectories. Through extensive experiments across diverse reasoning benchmarks, we show that MITS achieves substantial improvements over several strong baseline methods.

The remainder of this paper is organized as follows: Section 2 reviews background on tree search methods for LLM reasoning and current challenges, Section 3 presents our MITS framework in detail, Section 4 evaluates our approach on multiple reasoning benchmarks, Section 5 summarizes related work, and Section 6 gives conclusion, discusses broader implications and future directions.

## 2 Preliminary

**Tree-Search Based Reasoning.** To address reasoning tasks with large language models (LLMs), the problem is often framed as a multi-step generation process with chain-of-thought (CoT) prompting (Wei et al., 2022; Kojima et al., 2022), where the model decomposes and solves the problem step by step. However, CoT is restricted to a single reasoning trajectory, which limits exploration of the solution space. To overcome the limitation, more and more works adopt tree-based frameworks (Yao et al., 2023; Hao et al., 2023; Qi et al., 2025) that enable systematic exploration of multiple reasoning paths. Specifically, given a reasoning problem $q$, we construct a search tree $\mathcal{T}$ by producing and sampling the nodes step by step, where the root node corresponds to the question $q$ and each child node is an intermediate reasoning step $s$. A path from the root node $q$ to a node $s_n(n \geq 1)$ constitutes a candidate solution path $S = q \oplus s_1 \oplus s_2 \oplus \cdots \oplus s_n$, where the operator $\oplus$ denotes the sequential concatenation. Within the search tree $\mathcal{T}$, a set of solution paths $\mathbb{S} = \{S^1, S^2, \ldots, S^c\}(c \geq 1)$ is extracted, from which we aim to find the most plausible path as the final answer.

**Challenges in Tree Search.** Recently, within the tree-search paradigm (Yao et al., 2023) for LLM reasoning, Monte-Carlo Tree Search (MCTS) emerges as one of the most powerful and widely-used algorithms (Hao et al., 2023; Qi et al., 2025; Ding et al., 2024). However, evaluating the quality of intermediate reasoning steps in MCTS typically relies on repeated look-ahead rollouts to simulate future trajectories and back-propagate for value estimation, incurring substantial computational cost. In addition, prior work highlights the difficulty of defining a reliable metric for selecting the correct trajectory (Qi et al., 2025). To this end, we ask whether we can design a principled scoring function that can assess the quality of reasoning traces without simulated rollouts, thereby addressing both the computational efficiency and trajectory selection challenges simultaneously.

## 3 Methodology

This section introduces the proposed Mutual Information Tree Search (MITS) method, which constructs a search tree and evaluates reasoning path quality through an effective mutual information based criterion, without relying on looking-ahead simulations. As illustrated in Figure 1, there are three major components as follows.

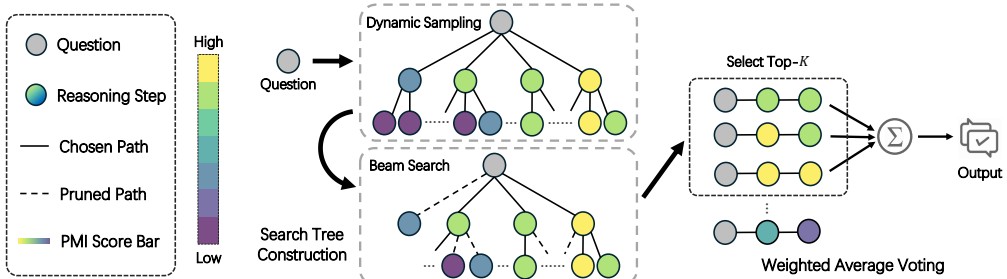

Figure 1: Overview for the MITS approach. The search tree is incrementally constructed by generating reasoning steps conditioned on prior context. Dynamic sampling controls the number of candidates (i.e., nodes) generated at each step. Beam search is applied to prune less promising paths. Weighted average voting is used to aggregate candidates and select the final output.

**1. Pointwise Mutual Information (PMI) Scoring.** Given a question and its reasoning path, our objective is to evaluate the path's quality. We employ Pointwise Mutual Information (PMI) as a principled and effective scoring metric for intermediate reasoning steps (section 3.1), which then serves as the criterion for guiding search tree expansion.

**2. Search Tree Construction.** Using PMI as guidance, we build the search tree with the beam search technique as demonstrated in section 3.2. Furthermore, we develop a dynamic sampling strategy, which allocates computation resources dynamically according to the uncertainty of the current step.

**3. Weighted Average Voting for Output Selection.** The search tree yields a set of candidate reasoning chains, from which we make the final prediction decision as in section 3.3. Unlike standard majority voting (Wang et al., 2023), which treats all predictions equally and overlooks their relative strengths, we propose a weighted average voting scheme that assigns weights to predictions according to their PMI scores, leading to more reliable answer selection.

### 3.1 POINTWISE MUTUAL INFORMATION (PMI) SCORING

An effective scoring function for reasoning tasks should not only favor solutions that appear plausible, but also quantify how much *specific* information a solution contributes to answering the question (Shi et al., 2024; Cui et al., 2025; Agarwal et al., 2025). In particular, the function should: (i) assign higher scores to solutions that are tightly coupled with the question, and (ii) penalize generic or spurious solutions that could apply broadly to unrelated queries. Mutual Information (MI) naturally satisfies these requirements, as it measures the reduction in uncertainty about solutions $S$ once questions $q$ are observed. Formally, the mutual information between questions and solutions is:

$$I(S; q) = \mathbb{E}_{p(S,q)} \left[ \log \frac{p(S, q)}{p(S)\, p(q)} \right], \tag{1}$$

where $p(S, q)$ is the joint probability of $q$ and $S$; $p(q)$ and $p(S)$ are the marginal probability of the question and reasoning path, respectively. While MI captures the average dependence between questions and reasoning paths, the quantity inside the expectation, $\log \frac{p(S,q)}{p(S)\, p(q)}$, evaluates the contribution of a particular question-reasoning pair. This term is known as the *pointwise mutual information (PMI)*. Since $\frac{p(S,q)}{p(q)} = p(S \mid q)$, the PMI score simplifies to

$$\mathrm{PMI}(q; S) = \log \frac{p(S \mid q)}{p(S)}. \tag{2}$$

Here, $\mathrm{PMI}(q; S)$ satisfies the aforementioned requirements for an effective scoring function, because (i) the probability $p(S \mid q)$ measures the degree of match for the reasoning process $S$ given the question $q$, and (ii) the denominator $p(S)$ penalizes generic paths that remain likely even without observing $q$.

**Probability Estimation.** To compute the PMI score, we need to estimate the probabilities $p(S \mid q)$ and $p(S)$ in Equation 2. Suppose a reasoning path $S$ is currently expanded up to the $n$-th step

$(n > 1)$, comprising a sequence of reasoning steps $(s_1, s_2, \ldots, s_n)$. The conditional probability $p(S \mid q)$ is decomposed as $p(S \mid q) = \prod_{i=1}^{n} p(s_i \mid q, s_1, \ldots, s_{i-1})$ and the marginal probability is $p(S) = \prod_{i=1}^{n} p(s_i \mid s_1, \ldots, s_{i-1})$. To estimate the conditional probabilities $p(s_i \mid q, s_1, \ldots, s_{i-1})$ and $p(s_i \mid s_1, \ldots, s_{i-1})$, we leverage the next-token prediction abilities of auto-regressive LLMs. Specifically, both probabilities are computed by multiplying token distributions within step $s_i$, with $p(s_i \mid q, s_1, \ldots, s_{i-1})$ conditioned on the full context including question $q$, and $p(s_i \mid s_1, \ldots, s_{i-1})$ conditioned only on previous reasoning steps. When $n = 1$, $p(S \mid q) = p(s_1 \mid q)$, and $p(S) = p(s_1)$. In implementation, to approximate $p(s_1)$, we prepend the special Begin-of-Sequence (`<bos>`) token to the language model.

**PMI Scoring with Incremental Update.** We are now able to calculate the PMI score given a complete question and reasoning chain. However, to extend the reasoning process step-by-step in a search tree, it is preferable to use an incremental update formula rather than re-computing PMI from the beginning at each step. Let $S_n = (s_1, s_2, \ldots, s_n)$ denote the reasoning path at step $n$, and $\text{PMI}_n$ be the PMI score up to the $n$-th step. We can have

$$\text{PMI}_n := \text{PMI}(q; S_n) = \sum_{i=1}^{n} \big[ \log p(s_i \mid q, s_1, \ldots, s_{i-1}) - \log p(s_i \mid s_1, \ldots, s_{i-1}) \big]. \quad (3)$$

The detailed derivation can be found in Appendix B.1. We transform it into an additive form using log probabilities for ease of computation. Then, the incremental update for extending the reasoning path from step $n$ to step $n+1$ is given by:

$$\text{PMI}_{n+1} = \text{PMI}_n + \big[ \log p(s_{n+1} \mid q, s_1, \ldots, s_n) - \log p(s_{n+1} \mid s_1, \ldots, s_n) \big], \quad (4)$$

with the initial condition $\text{PMI}_0 = 0$, i.e., the PMI score is zero before any reasoning step is generated. In this way, each update captures the incremental information gain contributed by the new step $s_{n+1}$ toward answering the question $q$, beyond what has already been accounted for by the preceding steps. This not only enables efficient computation during reasoning, but also provides an interpretable view of how individual steps contribute to problem-solving.

## 3.2 SEARCH TREE CONSTRUCTION

In this subsection, we introduce how to build the reasoning search tree $\mathcal{T}$ using the PMI criterion. We use a generator model $G$ to produce each step and an evaluator model $E$ to compute PMI scores. We first introduce how to generate each reasoning step, then the construction of the reasoning tree $\mathcal{T}$, which couples two mechanisms: (1) dynamic sampling that decides how many next-step candidates to propose; (2) PMI-guided beam search that evaluates and prunes candidates to retain only the most promising paths for further expansion. In short, dynamic sampling produces candidates, and beam search selects which of them advance to the next level of the tree.

**Reasoning Step Generation.** To construct the tree, a fundamental operation is to generate intermediate reasoning steps. At step $n + 1$, the generator model $G$ produces *multiple candidate steps* conditioned on the question $q$ and the partial reasoning path $(s_1, \ldots, s_n)$. As shown in Figure 2, we format generation with a CoT-style instruction (*e.g.*, "Let's think step by step") and delimit each step using a special marker (*e.g.*, `"Step"`), so that $G$ produces one step at a time before returning control for evaluation by evaluator model $E$. The first set of candidates is conditioned solely on $q$, while later candidates are conditioned on both $q$ and the preceding reasoning context.

> **Question:**
> Was ship that recovered Apollo 13 named after a World War II battle?
>
> **Input Prompt (Generating the 1st step):**
> Let's think this question step by step.
> Step1:
>
> **Input Prompt (Generating subsequent steps; e.g. the 3rd step):**
> Let's think this question step by step.
> Step1: Recovery Ship Identification. Apollo 13 splashed down in the South Pacific on April 17, 1970, and was recovered by the USS Iwo Jima (LPH-2).
> Step2: Ship's Namesake. The USS Iwo Jima (LPH-2) was explicitly named in honor of the Battle of Iwo Jima; a pivotal World War II engagement fought in February 1945.
> Step3:

Figure 2: Prompt examples for step-by-step reasoning generation, showing the iterative generation from initial step to subsequent steps, similar to (Lai et al., 2024).

**Dynamic Sampling via Entropy Quantification.** Through reasoning step generation, we can produce multiple candidate steps conditioned on the previous one. However, how many candidate steps to sample requires careful consideration to optimize test-time compute (Singhi et al., 2025; Snell et al., 2024). In particular, more computational resources should be allocated to reasoning branches with higher *uncertainty*, since they offer greater potential for generating and exploring diverse paths.

To this end, we use entropy to quantify uncertainty and sample next-step candidates dynamically. For a reasoning step $s_i$, its entropy is computed over the token distribution as:

$$H_i = -\sum_{v \in \mathcal{V}} p_i(v) \log p_i(v), \tag{5}$$

where $\mathcal{V}$ is the vocabulary and $p_i(v)$ denotes the probability of the token $v$ at step $i$. To adjust sampling adaptively, we divide the entropy into three intervals: high, moderate, and low. However, it is hard to use fixed thresholds to divide them, because entropy values can fluctuate given different tasks or different questions due to varying difficulty. To address this, we maintain a history of entropies for all generated steps $H_{1:m} = \{H_1, H_2, ..., H_m\}$, where $m$ is the total number of nodes in the current search tree, and use empirical *quantiles* to compute adaptive thresholds. Specifically, we set $H_{\text{low}} = \text{percentile}(H_{1:m}, 33\%)$ and $H_{\text{high}} = \text{percentile}(H_{1:m}, 67\%)$, which automatically partition entropy values into three regions: low ($H_i < H_{\text{low}}$), moderate ($H_{\text{low}} \leq H_i \leq H_{\text{high}}$), and high ($H_i > H_{\text{high}}$). Unlike fixed cutoffs, these quantile-based thresholds adapt to the varying difficulty across tasks and questions. Finally, the number of samples for step $i + 1$ is adjusted based on the entropy region. The final sampling number is set to $N_{\text{base}} + \Delta N_i$, where $N_{\text{base}}$ is the default sample size and $\Delta N_i$ is determined by the entropy level. The complete formulation for computing $\Delta N_i$ is given in Appendix B.2.

**Beam Search with PMI.** Once candidate steps are generated, the evaluator model $E$ computes their incremental PMI contributions, and updates the cumulative score to $\text{PMI}_{n+1}$ according to Equation 4. Since generating and evaluating all possible paths quickly becomes computationally expensive, we employ *beam search pruning* to ensure efficiency. Candidates are ranked by their cumulative PMI scores, and only the top-$B$ paths are retained for the next step, where $B$ denotes the beam width. This pruning keeps the tree computationally tractable while focusing exploration on the most informative reasoning directions. By tightly coupling candidate generation with PMI-based pruning, the method balances diversity and quality throughout the tree search.

## 3.3 Weighted Average Voting for Output Selection

After building the search tree $\mathcal{T}$, we need to select the final output from the collected reasoning chains $\mathbb{S} = \{S^1, S^2, \ldots, S^c\}$, where $c$ is the total number of candidate chains. A straightforward approach is to choose the chain with the highest PMI score. However, this can be brittle: the top-scoring chain may arise from spurious correlations between the query and particular reasoning steps (Zhao et al., 2024; Shao et al., 2025; Xie et al., 2025), or from overfitting to specific linguistic patterns (Wu et al., 2025; Li et al., 2025), leading to overconfident yet incorrect predictions. To mitigate this risk, we propose Weighted Average Voting, a simple but effective post-hoc operation by incorporating *prediction frequency* as a weight on PMI scores, as consensus among multiple reasoning paths often indicates more reliable conclusions.

Formally, given the reasoning chains, we first sort them according to PMI scores from high to low, and then select the top $K$ chains: $\mathbb{S}_{\text{top-}K} = \{S^1, S^2, \ldots, S^K\}$. For each chain $S \in \mathbb{S}_{\text{top-}K}$, we extract its final prediction $\text{Pred}(S)$. The frequency of each unique prediction is then computed as: $\text{Freq}(p) = |\{S \in \mathbb{S}_{\text{top-}K} : \text{Pred}(S) = p\}|$. After that, we can compute the frequency-weighted PMI score ($\text{PMI}^*$) for each reasoning chain:

$$\text{PMI}^*(q; S) = \text{PMI}(q; S) \cdot \frac{\text{Freq}(\text{Pred}(S))}{K}. \tag{6}$$

We re-rank the top $K$ chains according to $\text{PMI}^*(q; S)$ and select the chain with the highest one as our final prediction: $S^* = \arg\max_{S \in \mathbb{S}_{\text{top-}K}} \text{PMI}^*(q; S)$. This reweighting mechanism provides a regularization effect and achieves a balance between confidence and consensus.

Table 1: Accuracies of MITS compared with several baselines on different datasets. Columns are grouped by LLM backbones. Best results are in **bold**, and second best results are underlined.

| Method | QWEN2.5-3B | QWEN2.5-7B | PHI-3.5-MINI | PHI-4-MINI |
|---|---|---|---|---|
| | | *StrategyQA* | | |
| CoT | 47.34 | 65.47 | 54.47 | 51.25 |
| CoT-SC | 56.12 | 68.74 | 56.71 | 55.47 |
| ToT | 49.86 | 66.54 | 54.12 | 57.41 |
| RAP | 60.56 | 68.14 | 57.36 | 59.45 |
| rStar | 65.32 | 70.41 | 62.57 | 64.89 |
| MITS-F | 67.84 | 74.73 | 65.75 | **70.45** |
| MITS | **68.45** | **75.76** | **66.47** | 68.19 |
| | | *ARC-Challenge* | | |
| CoT | 69.11 | 71.50 | 67.52 | 72.25 |
| CoT-SC | 81.16 | 81.02 | 77.95 | 82.48 |
| ToT | 85.45 | 85.86 | 77.86 | 83.56 |
| RAP | 83.96 | 83.45 | 83.56 | 85.48 |
| rStar | 86.74 | 87.24 | 86.84 | 87.74 |
| MITS-F | 87.45 | **93.45** | 90.54 | **91.56** |
| MITS | **90.68** | 92.55 | **90.85** | 90.35 |
| | | *CommonsenseQA* | | |
| CoT | 66.67 | 66.50 | 55.86 | 65.71 |
| CoT-SC | 74.72 | 76.52 | 67.93 | 69.82 |
| ToT | 75.56 | 79.32 | 66.49 | 72.56 |
| RAP | 73.24 | 82.23 | 73.23 | 75.84 |
| rStar | 79.34 | 85.73 | 77.65 | 79.41 |
| MITS-F | **79.68** | **86.25** | **79.56** | **81.49** |
| MITS | 78.83 | 84.80 | 78.14 | 80.28 |

## 4 EXPERIMENTS

### 4.1 SETUP

**Backbone models and datasets.** MITS is a general algorithm and applicable to various LLMs and different reasoning tasks. We conduct main experiments on the LLMs QWEN2.5-3B-INSTRUCT, QWEN2.5-7B-INSTRUCT (Team, 2024), PHI-3.5-MINI-INSTRUCT and PHI-4-MINI-INSTRUCT (Abdin et al., 2024). All used models are instruction-tuned, and we omit the -Instruct suffix for brevity in the following sections and tables. We perform experiments on three datasets, StrategyQA (Geva et al., 2021) of questions that require implicit multi-hop strategies to answer, ARC-Challenge (Clark et al., 2018), which represents a scientific knowledge-intensive reasoning task, and CommonsenseQA (Talmor et al., 2019) of commonsense questions that require prior world knowledge to answer.

**Implementation details.** When calculating the PMI score (3), we observe that a single-step PMI value tends to be positive, *i.e.*, $p(s_{n+1} \mid q, s_1, \ldots, s_n) > p(s_{n+1} \mid s_1, \ldots, s_n)$, which induces a length bias whereby longer reasoning chains tend to yield larger PMI scores. To mitigate this effect, we normalize PMI through multiplying $\text{PMI}_n$ by factor $\frac{1}{n}$. In the stage of search tree construction, the maximum depth of the tree is set to 10, and the default sampling number $N_{\text{base}} = 3$. The evaluator model $E$ is set as same with the generator model $G$ by default. We set the beam width $B$ at 32 for the beam search process of MITS. We also run MITS-F, the variant algorithm of MITS by fully expanding the search tree without beam search, which allows us to analyze the trade-off between computational efficiency and reasoning performance. We set $K = 32$ for weighted majority voting for both of MITS and MITS-F. To guarantee reproducible inference process, we searched a set of hyperparameters, which is detailed in Appendix C.1.

## 4.2 MAIN RESULTS

**Baselines.** We compare MITS with three types of baselines: single CoT prompting, multiple CoT sampling and tree-search methods. Since MITS only leverages inference phase of language models, we focus on training-free baselines for fair comparison. **(1)** For single CoT prompting, we use CoT prompting (Wei et al., 2022; Kojima et al., 2022). **(2)** For multiple CoT sampling, we choose the widely adopted CoT with self-consistency method (CoT-SC; Wang et al., 2023), and employ majority voting to select the final prediction. We report the performances of sampling 32 times. **(3)** For tree-search methods, we include three strong baseline methods, Tree-of-thought (ToT; Yao et al., 2023), RAP (Hao et al., 2023) and rStar (Qi et al., 2025). We run ToT with Breath-First Search (BFS) to expand the tree. RAP and rStar utilize Monte-Carlo Tree Search algorithm (MCTS; Kocsis & Szepesvári, 2006) for tree search process, with 32 rollouts performed for each method.

**Performances on diverse reasoning datasets.** Here we present the performances of MITS against several baselines on diverse reasoning datasets in Table 1. We start by evaluating the effectiveness of MITS on general reasoning benchmarks. We highlight three observations. **(1)** MITS demonstrates substantial improvements over previous approaches across diverse tasks. For example, on StrategyQA dataset with QWEN2.5-3B, MITS achieves 68.45% accuracy compared to CoT's 47.34%, representing a remarkable +21.11% improvement. Similarly, on ARC-Challenge with QWEN2.5-7B, MITS reaches 92.55% accuracy, significantly outperforming the strongest baseline rStar by +5.31%. This substantial gain indicates that PMI-based scoring provides more effective guidance for reasoning path selection. **(2)** MITS consistently outperforms existing methods across different model scales and reasoning domains. Unlike baseline methods that exhibit varying effectiveness across different tasks, MITS maintains strong results across logical reasoning (StrategyQA), scientific reasoning (ARC-Challenge), and commonsense reasoning (CommonsenseQA). For instance, while RAP achieves competitive results on some tasks, it falls significantly behind MITS on StrategyQA (60.56% vs 68.45% on QWEN2.5-3B). **(3)** The comparison between MITS and MITS-F reveals the value of beam search in reasoning. While MITS-F shows improvements on CommonsenseQA, it generally underperforms MITS on StrategyQA (e.g., 74.73% vs 75.76% on QWEN2.5-7B). This indicates that PMI-based beam search can enhance performance by filtering out inferior reasoning paths early, preventing exploration from being derailed by low-quality branches.

**Computational efficiency comparison.** To validate the computational efficiency of MITS, we conduct experiments on the StrategyQA dataset using QWEN2.5-3B, comparing against CoT-SC, RAP, and rStar under identical inference settings. We measure wall-clock time from inference start to reasoning completion and report the average time that each method needs to solve a problem.

Table 2 presents the results, revealing significant efficiency advantages for MITS. While CoT-SC achieves the fastest time (2.75s), it delivers the lowest accuracy (56.12%). MCTS-based methods RAP and rStar require substantially more computation time (203.42s and 815.67s respectively, which are 3.2× and 12.7× slower than MITS) with limited accuracy improvements (60.56% and 65.32%). In contrast, MITS achieves the highest accuracy (68.45%) with moderate

Table 2: Compute efficiency comparison.

| Method | Time (s) | Acc. (%) |
|--------|----------|----------|
| CoT-SC | 2.75     | 56.12    |
| RAP    | 203.42   | 60.56    |
| rStar  | 815.67   | 65.32    |
| MITS   | 64.41    | 68.45    |

computational cost (64.41s), demonstrating the optimal accuracy-per-time trade-off among all methods. These results validate our claim that PMI-based scoring achieves superior reasoning performance with significantly reduced computational overhead compared to previous methods.

## 4.3 ABLATION STUDY

**Effectiveness under different number of rollouts.** MCTS-based baseline methods such as RAP and rStar construct the search tree through a rollout policy of MCTS algorithm. Increasing the number of rollouts expands the set of candidate solution trajectories and can lead to performance improvements, but it also incurs higher inference costs. Here we compare the accuracies of CoT-SC, RAP, rStar and MITS under different number of rollouts on the ARC-Challenge dataset as shown in Figure 3. For CoT-SC and MITS, we define number of rollouts same as the number of reasoning chains finally collected. We highlight two key findings: **(1)** With as few as two rollouts,

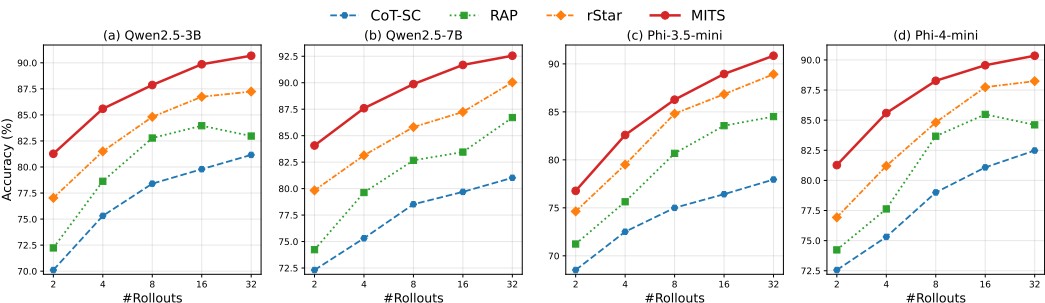

Figure 3: Performance comparison across different rollouts on the ARC-Challenge dataset. MITS not only demonstrates superior performance in early stages with fewer rollouts, but also continues to improve with increased computational budget.

Table 3: Results on ARC-Challenge with different pairs of generator and evaluator models.

| | Generator | |
| Evaluator | QWEN2.5-3B | QWEN2.5-7B |
| --- | --- | --- |
| QWEN2.5-1.5B | 86.78 | 87.20 |
| QWEN2.5-3B | 89.86 | 88.24 |
| QWEN2.5-7B | **90.61** | **91.68** |

Table 4: Results on the StrategyQA dataset with model QWEN2.5-3B and PHI-3.5-MINI using different aggregations for PMI scoring.

| | Models | |
| Aggregation | QWEN2.5-3B | PHI-3.5-MINI |
| --- | --- | --- |
| Sum | 65.89 | 63.26 |
| Average | **68.45** | **66.68** |

MITS achieves notable improvements in reasoning accuracy, underscoring the effectiveness of PMI-based scoring; **(2)** While rStar and CoT-SC exhibit continued performance gains with additional rollouts, RAP occasionally shows performance degradation as the number of rollouts increases. We hypothesize that this degradation stems from RAP's limited action space, which constrains the potential of its MCTS framework.

**Effectiveness of different evaluators.** As we coordinate two models generator $G$ and evaluator $E$ for MITS, how different evaluators $E$ impact the PMI score calculation process are under explored. Therefore, we conduct experiments on the ARC-Challenge dataset using Qwen series models as evaluators, with sizes ranging from 1.5B to 7B, as shown in Table 3. From the table we can observe that more powerful evaluator models usually yield better results. For example, take QWEN2.5-3B as the generator model, QWEN2.5-1.5B as evaluator can only has accuracy of 86.78%, while QWEN2.5-7B can have performance of 90.61%. This is likely because stronger evaluators provide more accurate probability estimates, leading to more reliable PMI values.

**Effectiveness of Weighted Average Voting.** We conduct the ablation study on the Weighted Average Voting technique introduced in Section 3.3, evaluating the impact of selecting different numbers of top-scoring reasoning paths $K$ for final answer selection. Using model QWEN2.5-3B, we run MITS with beam width of 32 across all three datasets and vary $K$ from 1 to 32. The corresponding performance results are reported in Table 5. The results show consistent performance improvements as $K$ increases, with the steepest gains observed at small $K$ values (*e.g.*, ARC-Challenge improves from 84.31% to 89.12% as $K$ increases from 1 to 8). Optimal $K$ values are dataset-dependent: StrategyQA peaks at $K = 16$ while ARC-Challenge and CommonsenseQA achieve best results at $K = 32$. Performance stabilizes as $K$ value becomes larger, indicating diminishing returns from incorporating additional reasoning paths.

**Ablation on different PMI variants.** In section 4.1, we mention to normalize PMI by averaging its accumulative value over the total steps to mitigate the length bias. Therefore, we conduct an ablation study comparing these two PMI aggregation approaches: (1) Sum aggregation, which accumulates PMI values across all reasoning steps, and (2) Average aggregation, which divides the cumulative PMI score by the number of steps. Table 4 presents results on the StrategyQA dataset with models QWEN2.5-3B and PHI-3.5-MINI. The results show that The average-aggregated (length-

Table 5: Performance across different values of top-$K$ for weighted majority voting with QWEN2.5-3B. Best results for each dataset are in **bold**.

| Dataset | Top-$K$ | | | | | |
|---|---|---|---|---|---|---|
| | 1 | 2 | 4 | 8 | 16 | 32 |
| StrategyQA | 63.34 | 65.41 | 67.44 | 68.04 | **68.86** | 68.23 |
| ARC-Challenge | 84.31 | 85.94 | 87.24 | 89.12 | 89.86 | **90.68** |
| CommonsenseQA | 74.78 | 75.16 | 76.64 | 77.02 | 78.49 | **78.83** |

normalized) PMI consistently outperforms the sum-based approach, with improvements of +2.56% on QWEN2.5-3B and +3.42% on PHI-3.5-MINI. This demonstrates that normalizing by path length prevents certain reasoning paths from being unfairly penalized, enabling fair comparison across paths of different lengths, which promotes the performance simultaneously.

## 5 RELATED WORK

**Prompting for LLM Reasoning.** Since the introduction of Chain-of-Thought prompting (Kojima et al., 2022; Wei et al., 2022), the field has witnessed a rapid development of prompting methods which focus on designing instructions and pipelines to elicit the potential reasoning capabilities within LLMs. Recent work make progress through several ways, such as decomposing a question into sub-questions to guide the reasoning process (Zhou et al., 2022; Khot et al., 2023; Yang et al., 2023) and demonstration selection (Zhang et al., 2023; Shum et al., 2023; Diao et al., 2024), which aims to select high-quality exemplars for better prompting performances. Higher-order thinking approaches, exemplified by abstract and analogical reasoning (Zheng et al., 2024; Hu et al., 2024a; Webb et al., 2023; Yasunaga et al., 2024; Yu et al., 2024), demonstrate effectiveness as well, compared to concrete demonstrations appended in prompts. Other approaches also include code-integrated prompting (Gao et al., 2023; Chen et al., 2023; Surís et al., 2023; Li et al., 2024; Zhou et al., 2024), which makes LLMs reason by coding, leading to a more structured way to express logical thinking processes. These methods aim to improve single-round inference performance and are orthogonal to ours.

**Tree Search for LLM Reasoning.** Recent advancements have shown that sampling diverse reasoning paths can significantly enhance performances compared to one-time greedy decoding in both theoretical and empirical ways (Brown et al., 2024; Snell et al., 2024), featured by Chain-of-Thought with Self-Consistency (CoT-SC) (Wang et al., 2023). Taking one step further, more work guide the reasoning process into tree structures (Yao et al., 2023; Xie et al., 2023), among which the most pioneering work is Tree-of-Thought (Yao et al., 2023). There are also works such as building search tree with beam search (Xie et al., 2023). Recently, Monte-Carlo Tree Search (MCTS) is regarded a powerful algorithm for advancing LLM reasoning, especially from the perspective of inference-time scaling (Hao et al., 2023; Qi et al., 2025; Ding et al., 2024). Moreover, tree structures are also deployed to help LLMs in planning tasks (Hu et al., 2024b; Gui et al., 2025).

## 6 CONCLUSION

In this work, we proposed Mutual Information Tree Search (MITS), a principled framework that leverages pointwise mutual information (PMI) to guide LLM reasoning without expensive look-ahead simulations. MITS introduces three key innovations: PMI-based scoring for reasoning path evaluation, entropy-based dynamic sampling for adaptive compute allocation, and weighted average voting for robust answer selection. Through comprehensive experiments across diverse reasoning datasets, MITS substantially outperforms several strong baseline methods by a large margin. These results demonstrate that information-theoretic principles can effectively steer LLM reasoning processes, achieving superior performances across diverse reasoning tasks while providing a computationally efficient algorithm to traditional tree search methods. Furthermore, we conduct extensive ablation studies, providing analysis and insights for more advanced LLM reasoning techniques.

## 7 ETHICS STATEMENT

All experiments use publicly released models, each operated strictly under its respective license and exclusively for research purposes. As detailed in Section 4.1, we assess model families with a focus on the Qwen (Yang et al., 2024; Team, 2024) and Phi (Abdin et al., 2024) series. Model performances are evaluated on widely adopted public datasets, including StrategyQA (Geva et al., 2021), ARC-Challenge (Clark et al., 2018), and CommonsenseQA (Talmor et al., 2019). We complied with all dataset and model usage terms; no personal data were collected or processed, no human-subjects research was conducted, and no personally identifiable information appears in the paper.

## 8 REPRODUCIBILITY STATEMENT

For reproducibility, we document the implementation and evaluation workflow in full. Section 3 outlines the end-to-end pipeline, with formal derivations and algorithmic details deferred to Appendix B.1 and Appendix B.2. Inference settings and key hyperparameters are compiled in Appendix C.1. Section 4.1 lists datasets, backbone models, and parameter scales. Upon acceptance, we will release a public repository containing code, configuration files, and scripts for running the full pipeline, together with instructions for data acquisition and environment setup.

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

# A  THE USE OF LARGE LANGUAGE MODELS (LLMS)

Large language models (LLMs), including Claude and ChatGPT, were employed only in a limited editorial role. Their use was restricted to grammar and syntax adjustments, improvements in clarity, stylistic harmonization, and occasional light rewording. They had no part in shaping research ideas, designing experiments, analyzing data, or producing technical material. All research contributions are the authors' own. The LLMs acted solely as language refinement aids, and every change was carefully checked and confirmed by the authors.

# B  ALGORITHM DETAILS IN MITS

## B.1  DETAILED DERIVATION FOR PMI SCORING

In section 3.1, we obtain the recurrent form for the calculation of PMI scores. Here in this appendix subsection, we derive and provide the calculation process for PMI score in recurrence relation, starting from Equation 2.

$$
\begin{aligned}
\mathrm{PMI}(q; S) &= \log \frac{p(S \mid q)}{p(S)} \\
&= \log \prod_{i=1}^{n} \frac{p(s_i \mid q, s_1, \ldots, s_{i-1})}{p(s_i \mid s_1, \ldots, s_{i-1})} \\
&= \sum_{i=1}^{n} \log \frac{p(s_i \mid q, s_1, \ldots, s_{i-1})}{p(s_i \mid s_1, \ldots, s_{i-1})} \\
&= \sum_{i=1}^{n} [\log p(s_i \mid q, s_1, \ldots, s_{i-1}) - \log p(s_i \mid s_1, \ldots, s_{i-1})].
\end{aligned}
$$

Therefore, we have the recurrence relation for PMI score calculation presented in Equation 3.

## B.2  FORMULA FOR DYNAMIC SAMPLING

In section 3.2, building upon the entropy-based partitioning described in the main text, we present the detailed mathematical formulation for computing the sampling adjustment $\Delta N_i$. The proportional control scheme adjusts sampling intensity based on how far the current entropy deviates from the moderate range:

$$
\Delta N_i = \begin{cases}
\min\left(2, \left\lfloor 2 \cdot V_p \cdot \dfrac{H_i - H_{\mathrm{high}}}{H_{\mathrm{high}} - H_{\mathrm{low}}} \right\rfloor\right), & H_i > H_{\mathrm{high}}, \\
0, & H_{\mathrm{low}} \le H_i \le H_{\mathrm{high}}, \\
\max\left(-2, -\left\lfloor 2 \cdot V_p \cdot \dfrac{H_{\mathrm{low}} - H_i}{H_{\mathrm{high}} - H_{\mathrm{low}}} \right\rfloor\right), & H_i < H_{\mathrm{low}}.
\end{cases} \tag{7}
$$

Here $V_p$ is the proportional gain parameter. The normalization factor $(H_{\mathrm{high}} - H_{\mathrm{low}})$ ensures that the adjustment magnitude is scaled relative to the entropy range, making the controller robust to different entropy scales. The final number of samples is computed with boundary constraints $N_i = N_{\mathrm{base}} + \Delta N_i$, where $N_{\mathrm{base}} = 3$ represents the base sample count. In practice, we enable dynamic sampling only when $m \ge 10$, ensuring that a sufficiently large number of samples has already been accumulated.

# C  IMPLEMENTATION DETAILS

## C.1  HYPERPARAMETERS

We conduct a comprehensive hyperparameter search to ensure stable and high-quality performances of the MITS algorithm. Here we provide the key parameters during the inference of LLMs. Specifically, we mainly record two groups of hyper-parameters, parameters for basic model inference and

parameters for search tree construction. As displayed in Table 6, for critical parameters of basic model inference, we search a set of parameters of temperature, top-k, and top-p for model decoding. For search tree construction, the sample number is set to 3, which means by default we sample 3 candidate next-steps given previous steps. The maximum depth of the search tree is 10. For generation of each step, we set the limit of 512 tokens. For beam width during the MITS process, we set it to 32.

Table 6: Hyperparameter Configuration for MITS.

| Parameter | Value |
|---|---|
| **Parameters for Basic Inference** | |
| temperature | 0.9 |
| top-k | 100 |
| top-p | 0.96 |
| **Parameters for Search Tree Construction** | |
| sample numbers | 3 |
| max depth | 10 |
| max new tokens | 512 |
| beam width | 32 |

