# OpenReview forum: "MITS: Enhanced Tree Search Reasoning for LLMs via Pointwise Mutual Information"
_ICLR.cc/2026/Conference — ICLR 2026 Conference Withdrawn Submission_

### Official Review · Reviewer_qVpg · 2025-10-31

**Soundness:** 2
**Presentation:** 2
**Contribution:** 2
**Rating:** 4
**Confidence:** 4

**Summary:**

This paper proposes an tree‑search‑based reasoning framework MITS for LLMs. It scores intermediate steps using pointwise mutual information by comparing their probabilities under question‑conditioned versus unconditioned contexts, then uses these scores to re‑rank and expand candidate paths via beam search. Using an entropy‑based policy, MITS allocates extra samples to uncertain nodes. Experiments report improved performance over baselines while maintaining computational efficiency.

**Strengths:**

1.Weighting generic and question‑conditioned reasoning steps is a principled way to estimate the quality of intermediate steps during tree‑search reasoning.
2.Adaptive compute allocation via an entropy‑based sampler focuses exploration on uncertain nodes, making the search more time‑efficient for a given accuracy target.

**Weaknesses:**

1.The accumulated step‑level PMI can be sensitive to the base model, decoding hyperparameters, and dataset domain. Although MITS applies basic sequence‑probability normalization, the paper does not thoroughly analyze robustness across models/prompts or control for step‑length bias in multi‑token steps. Moreover, while the paper claims to separate generic from question‑specific reasoning, it is not clear how stable that separation is across tasks.
2.The entropy may not be an effective symbol of the need of exploration.  Next‑token entropy does not always correlate with downstream correctness or path value in multi-step reasoning.
3.The PMI‑weighted voting scheme used to aggregate final answers is not theoretically justified. Without clear ablations across datasets and seeds, there’s a risk of overfitting to specific benchmarks.

**Questions:**

1.How sensitive are PMI rankings to decoding hyperparameters (temperature, top‑p), prompt format, and base model?
2.How well does MITS transfer across tasks (math, code, symbolic, planning) and models (small vs. large, instruction‑tuned vs. base)?
3.How does MITS aggregate the entropy in a reasoning step for adaptive sampling?
4.Did you compare entropy to alternative utilities (e.g., value functions, verifiers, or backward consistency) under matched compute?
5.Since the current voting mechanism lacks a solid theoretical foundation, have you investigated its re-ranking capability for the final answers?
6.Have you analyzed the distributional differences of PMI scores before and after re-ranking?
7.Although MITS improves performance, what kinds of errors does it actually help the model reduce? Or, conversely, what kinds of errors might it increase? Have you conducted any case analyses?
8.Under identical prompts and budgets, can you report wall‑clock time, total tokens, and model calls vs. baselines, plus a breakdown of MITS‑specific overheads?
9.How effective is average PMI over tokens at removing residual biases from tokenization and segmentation?
10.Do you cache “conditioned” and “unconditioned” probabilities for calculating P(step) during generation?

---

### Official Review · Reviewer_HAHf · 2025-11-02

**Soundness:** 2
**Presentation:** 2
**Contribution:** 2
**Rating:** 4
**Confidence:** 4

**Summary:**

This paper presents the Mutual Information Tree Search (MITS) framework, designed to enhance reasoning in large language models (LLMs) by leveraging pointwise mutual information (PMI) to evaluate the quality of reasoning paths during inference. MITS aims to improve computational efficiency compared to existing tree search methods like Monte Carlo Tree Search (MCTS) by using PMI to guide the expansion of the search tree. The paper also introduces dynamic sampling and a weighted voting mechanism for final output selection, showing improvements on several reasoning benchmarks.

**Strengths:**

- Computational Efficiency: The proposed method reduces computational overhead significantly compared to traditional exhaustive search methods (such as MCTS), which is valuable for real-world applications where time constraints are crucial.
- Results: The paper reports competitive results on diverse reasoning tasks (e.g., StrategyQA, ARC-Challenge) compared to existing baseline methods, which indicates the effectiveness of the approach.

**Weaknesses:**

- Lack of Clear Novelty and Motivation: The novelty of the idea and the core insight behind PMI-based scoring is not sufficiently emphasized. While PMI as a scoring mechanism is interesting, its motivation and the specific challenges it addresses need to be presented more clearly. The paper could benefit from a more detailed explanation of why PMI, in particular, offers significant advantages over other information-theoretic or heuristic measures.
- Outdated Baseline Models: The use of the QWEN2.5 series as the backbone model in the experiments is somewhat outdated, especially with the release of newer versions like QWEN3. These models are likely to provide more accurate results and could help better evaluate the potential of MITS. An experiment using the latest models would strengthen the paper.
- Limited Experimental Validation: While the paper provides comparisons with baseline methods, the experiments lack depth in some areas. For example, the evaluation on QWEN2.5 could be extended to more recent models or larger-scale tasks to demonstrate the robustness and scalability of the approach. Moreover, a more thorough ablation study could shed light on the contribution of each individual component (PMI scoring, dynamic sampling, etc.).

**Questions:**

1.	The paper mentions that PMI is used to evaluate reasoning paths during the inference process. Could you provide more insights into how PMI improves upon traditional methods in terms of capturing problem-specific reasoning?
2.	The results seem to be based on older versions of the QWEN model (QWEN2.5). Have you considered evaluating MITS with the latest models, such as QWEN3, to provide a more current assessment of its performance?
3.	Could you provide more detailed experiments or comparisons that test the scalability of MITS with larger models and more complex reasoning tasks? This would help assess its robustness in real-world applications.

---

### Official Review · Reviewer_SMA6 · 2025-11-02

**Soundness:** 3
**Presentation:** 3
**Contribution:** 2
**Rating:** 4
**Confidence:** 4

**Summary:**

In this paper, the authors propose a tree-search inference framework which they term is as MITS, that scores intermediate reasoning steps with pointwise mutual information, then expands and prunes the search tree accordingly, and finally aggregates candidate answers via weighted average voting. Particularly, a tree is constructed by generating multiple next-step candidates. The number of samples per node is set dynamically using token-entropy quantiles to allocate more compute to uncertain branches. Rather than plain majority voting, the method reweights top-K chains. Experimental results across StrategyQA, ARC-Challenge, and CommonsenseQA show improvements over CoT, CoT-SC, ToT, RAP, and rStar.

The authors claim their contributions as three-fold: `` a principled scoring approach for intermediate reasoning steps using Pointwise Mutual Information``, `` an entropy-based dynamic sampling strategy that adaptively allocates computational resources according to the uncertainty at each reasoning step``, and `` a weighted voting scheme that combines PMI scores with prediction consensus ``. These claims raise some of my concerns, which would be detailed in the following sections.

**Strengths:**

``S1``: The paper is well written. In general, PMI offers a clear, information-theoretic criterion for intermediate step quality rather than relying on long rollouts or ad-hoc heuristics.

``S2``: The proposed methodology such as the entropy-quantile dynamic sampling that allocates budget to uncertain fronts, also coupled with PMI-based beam pruning, makes sense.

**Weaknesses:**

``W1``: The paper is based on MCTS-tree-search-based reasoning paradigm. And as such, the paper benchmarks only CoT-SC, ToT/MCTS-style methods. I would have expected to see some comparisons against models trained with GRPO or similar RL paradigms (e.g., DeepSeek-R1/R1-Zero), which in fact currently define the state of the art for general-purpose reasoning. I clarify that MCTS-based reasoning is also a significant research domain, but the authors should mark the scope on what scenario each paradigm is good at.

``W2``: The method does not appear to use or compare to Process Reward Models (i.e. PRMs) or other step-level verifiers as a scoring alternative for step quality. I also fail to find a PRM discussion in the method and related-work sections.

``W3``: One of the main motivations of this paper is reducing the computational burden of tree exploration. However, this domain has been studied with many works, which are not compared or sufficiently discussed.

``W4``: In Tab. 1, the authors are suggested to include a reference for each comparative method, with additional inclusion of the above aforementioned critical methods for further comparison. In Tab. 2, the authors are suggested to include the detailed GPU configurations.

**Questions:**

``Q1``: A PRM-based step scorer (process supervision) as a plug-in comparator or hybrid with PMI?

``Q2``: It would be great and appreciated if the authors could discuss: could MITS close the gap against GRPO-trained reasoning models at similar inference budgets?

---

### Official Review · Reviewer_1sou · 2025-11-03

**Soundness:** 2
**Presentation:** 3
**Contribution:** 2
**Rating:** 2
**Confidence:** 4

**Summary:**

This paper introduces a prompting framework for reasoning called Mutual Information Tree Search (MITS). The main idea is to find the optimal path within a tree search for LLMs by integrating mutual information and entropy.
The authors propose a scoring mechanism that quantifies the informativeness and specificity of intermediate reasoning steps. Then, they combine it with entropy-based dynamic sampling to allocate computation to uncertain nodes and a weighted voting mechanism for final answer aggregation.

**Strengths:**

The motivation is clear. The paper aims to efficiently enhance the evaluation and prioritization of reasoning paths without expensive rollouts in inference-time reasoning. The target has a lot to explore.

Precise figures and tables for readers to understand.

**Weaknesses:**

For Related Work and Introduction:

i. The citation and discussion of related works are limited. There already exist many studies on optimal path selection in multi-path reasoning and on adopting mutual information and entropy in reasoning optimization. The authors do not clearly introduce and compare their work to these prior works, or explain the novelty and insights.

ii. The whole work is based on intuitive hypotheses, like high-quality reasoning paths should exhibit high mutual information with the question. These hypotheses are not empirically validated. If these ideas originate from prior research, the corresponding references should be clearly acknowledged. And the connections to this work should be explicitly discussed.

For Experiments:

i. The paper does not specify how they compute the PMI on LLMs, and lacks the details of how they normalize or handle token dependencies.

ii. The independent contributions of each module (PMI scoring, entropy sampling, weighted voting) are not explicitly verified.

iii. Section 4.2, the Computational Efficiency Comparison, this part lacks sufficient details about the experimental setup and measurement protocol. The reported times can barely be explained or compared, as the baseline settings and testing conditions are unclear.

**Questions:**

See above.

And since the authors mentioned several tree-search methods in their related work part, why these works were not compared?

---

### Note · Authors · 2025-11-25

I have read and agree with the venue's withdrawal policy on behalf of myself and my co-authors.